# The Impact of Symbolic Representations on In-context Learning for Few-shot Reasoning

**Hanlin Zhang**[1,*] **Yi-Fan Zhang**[2,*] **Li Erran Li**[3] **Eric Xing**[1,4,5]
[1]Carnegie Mellon University, [2]Chinese Academy of Science, [3]AWS AI, Amazon,
[4]Mohamed Bin Zayed University of Artificial Intelligence, [5]Petuum Inc.

## Abstract

Pre-trained language models (LMs) have shown remarkable reasoning performance using explanations (or "chain-of-thought" (CoT)) for in-context learning. On the other hand, those reasoning tasks are usually presumed to be more approachable for symbolic programming. To make progress towards understanding in-context learning, we curate synthetic datasets containing equivalent (natural, symbolic) data pairs, where symbolic examples contain first-order logic rules and predicates from knowledge bases (KBs). Then we revisit neuro-symbolic approaches and design a model LMLP that learns from demonstrations containing logic rules and corresponding examples to iteratively reason over KBs, recovering Prolog's backward chaining algorithm. Comprehensive experiments are included to systematically compare LMLP with CoT in deductive and inductive reasoning settings, showing that LMLP enjoys much better length generalization even with substantially less parameters.

## 1 Introduction

There are emerging interests in leveraging LMs to enable planning [23, 15], heuristic search [11] and symbolic inference [49, 55, 56]. Among them, *chain of thought* prompting [49] shows that taking `(input, explanation, output)` as in-context examples for LMs can lead to significant performance gain in reasoning tasks. However, like most fine-tuned LMs, generalizing compositionally is still challenging [58], which means that it struggles to reuse knowledge for solving harder problems with unseen combinations [22, 1, 19].

One notable case is that LMs would suffer from catastrophic performance degradation when tested on sequences longer than training ones (Figure 1). As a solution, *least-to-most prompting* [58] takes inspirations from neuro-symbolic programs and proposes to tackle the challenge by modularized prompting on the reduced problem. This divide-and-conquer strategy provides powerful tools to improve LMs' reasoning, but also poses an additional challenge: what is the right representations for in-context samples? How does natural language explanation compared with symbolic provenance when acting as prompts?

Our goal is to answer these questions by comparing the natural and symbolic paradigms closely. To provide insights into in-context learning for reasoning tasks from a symbolic perspective, we study relational reasoning over both natural language and KBs. This is because of the reliability and controllability of KBs:[2] given a query, it is easy to find a reasoning path in KBs as provenance without hand-crafting rationales [59, 49, 55] as part of the prompts. To few-shot learn from symbolic demonstrations and plan simultaneously in an explainable and scalable manner, we propose Language Models as Logic Programmers (LMLP). LMLP uses logic rule templates and examples combined

36th Conference on Neural Information Processing Systems (NeurIPS 2022).

---

*Equal contribution

[2]KBs are mostly constructed with clear pipelines and strong supervisions.

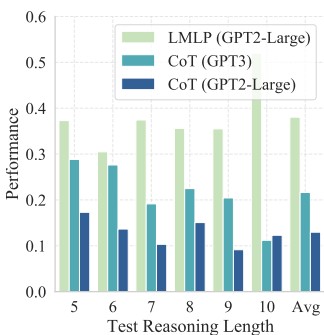

Figure 1: Relational reasoning performance (human evaluation accuracy) comparisons on the CLUTRR-LP given training data with story length 2, 3, 4.

with pre-trained knowledge to do in-context learning iteratively for answering a relational query. Specifically, given a goal query as the in-context example that can be interpreted as a question answering (QA) task, we search or retrieve a related task example with a corresponding logic rule (Figure 2). Then we concatenate the context and task description as the input of an autoregressive planning LM. At each step of generation, we use a masked translation LM to compare the similarity between the generated natural language sentences and encoded `(subject, relation, object)` predicates in the KB. Therefore, each generated sentence is transformed into a top-k similar predicate and the reasoning path is confined within the KB. The process is iterated until a predefined maximum iteration or the target of interest is reached (Figure 3) and the generated reasoning path is evaluated manually.

To evaluate the reasoning capability of CoT and LMLP, we curate two datasets and design a series of experiments, aiming to compare two recent in-context learning paradigms and explore both symbolic and naturalistic scenarios. Specifically, we adopt synthetic datasets containing `(natural, symbolic)` data pairs. The symbolic part contains predicates and first-order logic (FOL) rules which are well-suited for investigating the role of symbolic representations for few-shot reasoning. And the natural part includes a natural language story with the same set of entities and relations. Moreover, we create experimental settings that are infavorable for LMLP since (i) we use GPT-2 and SentenceBERT as its backbones, which is known to be of **much smaller scale** compared to CoT which is usually based on GPT-3 [6] or PaLM [9]; (ii) LMs are pre-trained over natual language sentences as opposed to KBs, which creates substantial gaps in semantics and representations, thus poses a **grounding** challenge where LMs are known to be ineffective [3].

Through conrolled experiments on relational reasoning, we find that (i) CoT prompting still struggles to solve compositionality challenge. On the other hand, as reasoning length increases, even under the above unfair settings, LMLP can still work much more reliably by taking symbolic inputs (predicates) to explicitly separate logic and control for improved problem-solving [21]. (ii) It is a common belief that large pre-trained LMs learned via imitation are not grounded in contexts that required learning with rich experiences. However, experimental results show that eliciting LMs with logic rules and examples using in-context learning, which maps the conceptual structure of the space learned from text onto a new structured space, is sufficient for solving some challenging reasoning tasks over KBs. (iii) LMs cannot solve relational reasoning tasks effectively if proper demonstrations containing target relation and correct input-label mappings are not provided, which is complementary to evidence in in-context examples which are poorly understood and manifest many intricate design choices [57, 25, 29].

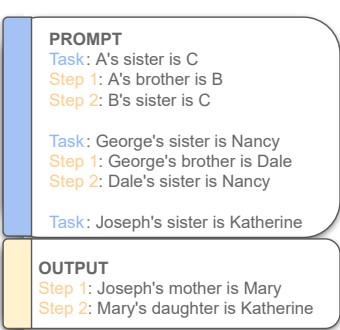

Figure 2: A deductive reasoning example for LMLP. LMLP retrieves a first-order logic rule and its corresponding grounded example to answer the target task. The reasoning path explains the concept of *sister*.

## 2 Related Works

**Neuro-symbolic reasoning** approaches are proposed for logic programming in statistical relational learning (SRL) but they usually have several key limitations. General ILP (Inductive Logic Programming) problem involves several steps that are NP-hard: (i) the rule search space grows exponentially in the length of the rule; (ii) assigning the logic variables to be shared by predicates grows exponentially in the number of arguments, which we refer as variable binding problem; (iii) the number of rule instantiations needed for formula evaluation grows exponentially in the size of data. A fundamental challenge in logic programming is to separate logic and control for improved efficient problem-solving [21]: with a determined control or theorem-proving strategy, the remaining process is to specify what is to be done via logic programming. Following the principle, traditional neuro-symbolic algorithms like backward chaining [43] and its continuous relaxation [30, 31, 56] are proposed, where unification between terms is replaced by the similarity between their embedding representations. We refer readers to Appendix A for more details and emphasize that those hybrid approaches can be inefficient and ineffective. Instead, with LMs, one can model the distribution of optimal decisions, and generate suitable candidate steps to proceed. Therefore, LMs offer a generic way of modeling the output space and executing classic symbolic algorithms.

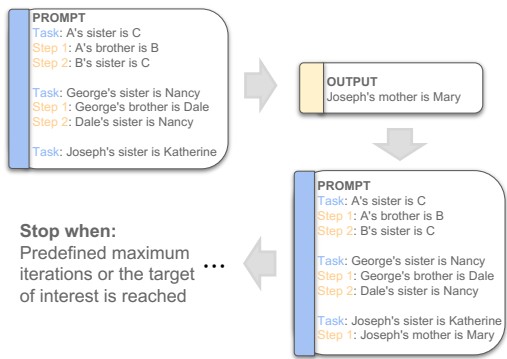

Figure 3: Illustration of step-wise prompting of LMLP.

**In-context learning** concerns feeding input texts describing a task with some examples to the black-box model for learning the task [6]. Many works show that there are intricate design choices like prompt formats [16, 25, 57, 29], example choices and their ordering [57, 27], pre-training data distribution [52, 45, 7] and model architectures [7] to improve the LMs' powerful and versatile in-context learning ability. There are recent works focusing on bootstrapping LMs with natural language explanations, intermediate steps, or rationales for reasoning [59, 49, 33, 55]. Recent works showcase both some positive [10] and negative results [17, 14, 47] in adapting LMs for symbolic or logical reasoning. Though there are some encouraging progress [10, 49, 9, 55], they require a significant amount of computation for re-training and human annotations about reasoning paths or explanations [59, 49]. Moreover, their entangled nature with natural language make them hard to make robust inference over symbolic factual knowledge. However, our goal is fundamentally different from theirs in investigating the role of symbolic representations on few-shot reasoning using in-context learning. Moreover, related works typically *finetune* the model using rationales or explanations [59, 55] or focus on natural language based reasoning such as commonsense reasoning, arithmetic reasoning, open-domain question answering [49], concept grounding [34] etc. Synthetic ontology datasets are constructed in [44] to understand the failure modes of CoT reasoning but they are in natural language forms instead of investigating the reasoning done over interpretable symbolic structures as we do. Therefore, all the above works are substantially different from our goal of exploring the representations of prompts in-context learning.

## 3 Methodology Overview

We consider the reasoning task with an SRL query as the question we are interested in with some background knowledge as the context. The relational information in the query and context can be

expressed either using natural language or a `(subject, relation, object)` predicate/triplet. There is a KB with facts $\mathcal{F}$ and (FOL) rules $\mathcal{R}$ to support the QA above. There are two ways for representing the problem - symbolic or natural language, which leads to the designs below.

**Datasets construction.** As shown in Table 1, we need to curate new symbolic datasets from the original ones into (i) *A query subset* containing predicates needed for proving. (ii) *A set of facts* $\mathcal{F}$ containing all the available facts/predicates, which composes a KB, and (iii) *A set of rules* $\mathcal{R}$ containing examples (A task and its proofs) extracted from the training set using backward chaining based neuro-symbolic reasoner [42]. See appendix C.2 for more details.

| Dataset | Natural Language Samples | Query | Facts $\mathcal{F}$ | Logic rules $\mathcal{R}$ |
|---|---|---|---|---|
| CLUTRR | Task: What's the relationship between Ashley and Nicholas? Story: Ashley told her daughter Lillian to wash up. Dinner was ready. Lillian called her brother, Nicholas up to see how he was doing after surgery. | (Ashley, son, Nicholas) | (Ashley, daughter, Lillian) (Lillian, brother, Nicholas) ... | Task: Ashley's son is Nicholas Step 1: Ashley's daughter is Lillian Step 2: Lillian's brother is Nicholas |
| Countries | Task: Is palau located in oceania? | (palau, locatedIn, oceania) | (palau, locatedIn, micronesia) ... | Task: palau locatedIn oceania Step 1: palau locatedIn micronesia Step 2: micronesia locatedIn oceania |

Table 1: Examples of data processing and curation.

**Language Models as Logic Programmers.** In LMLP, given a query `Task: Joseph's sister is Katherine`, which consists of two entities `Joseph, Katherine` and a target relation `sisiter`. Our task is to find a proof path from `Joseph` to `Katherine` where the relationship `sisiter` can be correctly inferred. On a high level, LMLP leverages an abstract logic rule `Sister(A,C) ← Brother(A,B) ∧ Sister(B,C)` and its grounded example `Sister(George, Nancy) ← Brother(George, Dale) ∧ Sister(Dale, Nancy)` to derive the answer for the query `Sister(Joseph, Katherine)` (Figure 4(a)).

To achieve this goal using in-context learning, at first, examples and logic rules in $\mathcal{R}$ are selected. For example, in Figure 3, LMLP samples one logic rule and its grounded example, which is concatenated with the query `Task: Joseph's sister is Katherine` as a prompt for the planning LM. LMLP samples a logic rule with the same target relation (e.g., `sister` in the above example) but different entities.[3] We then repeatedly generate contents by prompting an autoregressive planning LM $\mathcal{P}_\theta$ (Figure 3): The generated output is converted to particular predicates in the KB using the similarity of the embedding from a translation LM $\mathcal{T}_\phi$, implemented as a sentence-specific Masked LM. By constraining the output space of $\mathcal{P}_\theta$ with an external KB this way, LMLP is expected to produce more plausible provenance for explaining the reasoning process of final prediction. To improve coherency, we enforce the chain rule transition constraints: the tail entity of the previous predicate should be the same as the head entity of the next predicate for each output step. The model terminates when predefined maximum iterations or the target of interest is reached. The faithfulness of the reasoning path is governed by post-hoc human evaluations. The overall algorithm is described in Algorithm 1 in Appendix B.

Using prompting supported by KBs, we bootstrap the reasoning process from LMs in a few-shot manner (Figure 3). This is in stark contrast to popular methods that need expensive human annotations and retraining [59, 48, 15, 49, 55] or uncontrollable using only pre-trained knowledge [20].

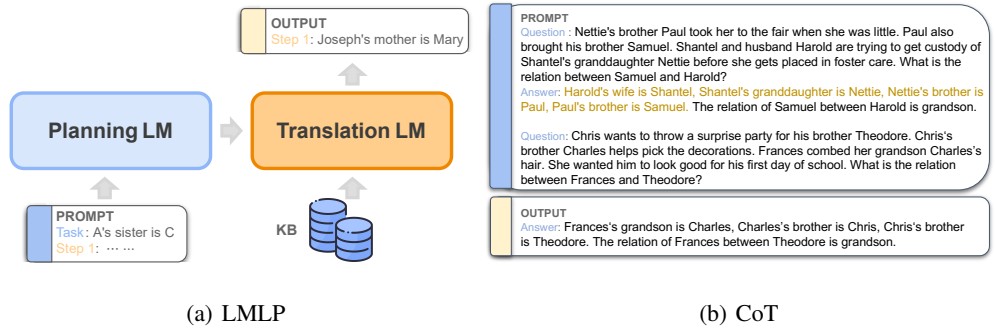

(a) LMLP                (b) CoT

Figure 4: Schematic overview of (a) LMLP and (b) CoT.

---

[3]We compare other sampling strategies in Table 2

**Chain-of-Thought prompting.** CoT [49] solves complicated multi-step reasoning tasks by providing explanations, which is also intuitive for our multi-hop SRL tasks since we can take intermediate reasoning paths as explanations.

Figure 4(b) shows an example of applying CoT to solve an SRL task from the CLUTRR dataset [46]: given an in-context sample in the form of `(input, explanation, output)`. LMs are expected to imitate the reasoning process of the given explanation to generalize to a new query. The explanation of each question is generated just the same as the rules set $\mathcal{R}$, which extracted from the training set using a neuro-symbolic reasoners and converted to natural language forms. Specifically, the in-context exemplar adapts LMs to another sample containing multiple relations and a query for the relation between two entities *"What is the relation between Theodore and Frances?"*, CoT first generates a reasoning path from *Frances* to *Theodore*, namely *"France's grandson is Charles, ..., Chris's brother is Theodore."*, and finally answers the query: *"The relation of Frances between Theodore is grandson"*. With such a prompt, LMs are expected to generate both the reasoning paths and the resulting queried relation. Note that the explanation in CoT is extracted from the story in the question, which contains much clearer information than the logic rules for LMLP.

# 4   Experiments

We now describe the experimental setups, empirically evaluate LMLP and compare it with existing methods. See Appendix C for full details of data preprocessing and performance evaluation.

**Settings.** We curate two datasets for evaluating the in-context learning capability of LMs for reasoning: CLUTRR-LP and Countries-LP, which are based on CLUTRR [46] and Countries [4] datasets respectively. CLUTRR [46] contains a group of KBs, where each node denotes a family member and edges are family relations. The target of CLUTRR dataset is to infer a two-family members' relationship that is not explicitly mentioned. The training set of CLUTRR consists of graphs that the target relation can be inferred by traversing a limited number of edges while the relation in the test set needs more traversing steps for inference, which allows controlled studies on compositionality. Another intriguing property of CLUTRR is that there are ground truth **one-to-one correspondances** between KBs and natural language stories, which exactly suits our needs. Countries [4] concerns link prediction, where countries, regions, and sub-regions are entities and relations containing *LocatedIn* and *NeighborOf*. Countries has three tasks, *R1,R2*, and *R3* based on three different data partitions strategies according to [42].

Table 2: Numerical results and ablation on the length of test samples on CLUTRR-LP.

| Planner | No Prompt | Only Rule | Random | Entity-based | CoT | LMLP-reverse | LMLP |
|---------|-----------|-----------|--------|--------------|-----|--------------|------|
| 0.0973 | 0.1514 | 0.1622 | 0.2919 | 0.2000 | 0.173 | **0.3730** | 0.3297 |
| 0.1810 | 0.1238 | 0.1524 | 0.2095 | 0.1429 | 0.1365 | **0.3048** | 0.2476 |
| 0.2258 | 0.2000 | 0.2129 | 0.2323 | 0.1742 | 0.1032 | **0.3742** | 0.2581 |
| 0.1037 | 0.2222 | 0.2000 | 0.3111 | 0.2370 | 0.1506 | **0.3556** | **0.3556** |
| 0.1048 | 0.1935 | 0.2177 | 0.1613 | 0.1855 | 0.0914 | **0.3548** | 0.2984 |
| 0.1230 | 0.2869 | 0.2131 | 0.3934 | 0.2705 | 0.123 | **0.5246** | 0.4754 |

# 5   Concluding Remarks

We systematically explores in-context learning of LMs through a symbolic reasoning perspective, showing that LMs can be prompted with logical demonstrations to generate plausible provenance for reasoning tasks over KBs. The empirical superiority of LMLP provides fresh insights towards understanding in-context learning suggests a way to ground GPT that types language to non-linguistic symbols in the KB. Moreover, few-shot in-context learning with LMs provides a convenient way to incorporate background knowledge without re-training. As its implications, it supports interpretable multi-hop reasoning and easy integration of domain knowledge that are key desiderata of neuro-symbolic approaches. We hope our empirical study can inspire more explorations on the trade-off between natural language and programming language for effective adaptation and planning of LMs.

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

# Appendix

## A Extended Related Work

**Neuro-Symbolic Reasoning.** ILP [32] and its neural version [54] are unable to reason about disjoint relations in confront of missing links when KBs are noisy like in FreeBase, which means ILP only synthesizes rules based on existing relations. Methods like Neural-LP [53] and RNNLogic [37] require enumeration of all possible rules given a max rule length T. Thus the complexity of these models grows exponentially as maximum rule length increases, which is an significant disadvantage for systematicity problems. For deductive reasoning, NTP [42] and its improved versions [30, 31] require hand-crafted templates to imitate backward chaining for deductive reasoning. This belies the considerable user burden of authoring the templates which then fundamentally biases the tool toward a specific subset of programs that the author has in mind. Moreover, the performance and efficiency of NTP is far from satisfactory: the performance usually lags far behind its neural counterparts like knowledge graph embedding methods [24]; during both training and inference, NTPs need to compute all possible proof trees needed for proving a query, relying on the continuous unification of the query with all the rules and facts in the KB. The search space of existing works is exponentially large, which makes them hard to scale up in general [30, 8].

**LMs for Theorem Proving.** Most works focus on proving formal mathematical theorems: GPT-f [36] shows promising results by generative language modeling over mathematical formulas. Systematicity of LMs when training on proofs is evaluated in [13] but shows negative results in generalizing to unseen proof steps in extrapolation and complex language. Three synthetic tasks inspired by three reasoning primitives of deduction, induction, and abduction are demonstrated in [51]. The above works provide insights into understanding LMs' reasoning capabilities. Though they share similar problem structures like compositionality with ours, they fundamentally require large-scale pre-training and fine-tuning due to the mismatch between Wikipedia pre-training corpora and mathematical formulas. Such a re-training requirement not only results in computational inefficiency but lacking in compositional generalization to longer proof steps unseen during training [13].

**Symbolic Reasoning with LMs.** Large LMs pre-trained on open-domain text corpora have achieved impressive advances in natural language generation and understanding tasks [18, 6]. By self-supervised imitation on human-generated texts, LMs contain rich factual knowledge [35, 5, 41] and linguistic structures [28], serving as a versatile inference regime for various downstream tasks [6, 26]. Among them, GPT-3 stands out by its few-shot generalization to unseen cases without further fine-tuning given in-context samples as demonstrations [6]. However, it is a common belief that LMs have not yet enjoyed a comparable success in tasks that require extensive planning and grounding [12, 2, 3] as well as symbolic reasoning [17, 14, 39].

## B Algorithm Description

Algorithm 1 describes the procedure or LMLP. It can also be illustrated in Figure 4(a).

## C Additional Experimental Setups and Results

**Implementation details.** For LMLP, we implement the planning LM $\mathcal{P}_\theta$ as GPT-2 [38], the translation LM $\mathcal{T}_\phi$ as Sentence BERT (Sent-BERT) [40] based on Hugging Face Transformers [50]. The default model for Translation LM is Sentence-RoBERTa-Large and for Planning LM is GPT2-Large [38] pretrained on large corpora. For CoT, we follow the original paper [49] to sample in-context samples and use GPT-3 (text-davinci-002) which is accessed using OpenAI API for implementation. We conduct all the experiments on a machine with four Nvidia TITAN XP (10GB) GPU cards.

Since prompt formats lead to significant performance variations [25], we propose to explore two simple design choices for LMLP and find that they can further boost the reasoning capacity. (i) Multiple examples for prompting. Denote $N$ the number of examples we used in one proof task. Table 9 shows two examples with $N = 1$ and $N = 2$ are supplied respectively. The intuition is that, getting more examples in the prompt can make LMs better recognize the proof task and thus produce more reliable reasoning paths. See the experimental section for empirical verification. (ii)

---
**Algorithm 1** Generate proof path from Pre-Trained Language Models.
---
**Require:** Planning LM $\mathcal{P}_\theta$, Translation LM $\mathcal{T}_\phi$, Query set $\mathcal{Q}$ that contains all query triplets, $\mathcal{F}$ that contains all available facts, $\mathcal{R}$ that contains all the available rules or proof examples.
    **for** $(s, r, o) \in \mathcal{Q}$ **do** // *s, r, o denote subject entity, relation and object entity respectively.*
        Find $e \in \mathcal{F}$, whose task relation is $r$. Convert $e$ to the prompt $e'$.
        **while** Max step is not reached **do**
            Sample 10 sentences $\{x_i\}_{i=1}^{10}$ from $\mathcal{P}_\theta$ with current prompt.
            Set $\mathcal{F}' \in \mathcal{F}$ whose first entity is $s$.
            **if** $|\mathcal{F}'| == 0$ **then**
                Break    // *No available facts in the $\mathcal{F}$ start with entity s.*
            **for** $x \in \{x_i\}_{i=1}^{10}$ **do**
                $score_i = \max cosine(\mathcal{T}_\phi(x, r)); \forall r \in \mathcal{F}'$// *Calculate cosine similarities of s to facts in* $\mathcal{F}'$.

                $idx = \arg\max cosine(\mathcal{T}_\phi(x, r)); \forall r \in \mathcal{F}'$// *Select $r \in \mathcal{F}'$ with the highest similarity to* $x$.

                $x' = \mathcal{F}'[idx]$
            Choose the highest score rule $x' = (s, r', o')$ as the next proof step and append it to the prompt.
            **if** $o' == o$ **then**
                Break    // *The object entity converges to the target entity o.*
---

Table 3: Qualitative comparison of CoT and LMLP over the same example on CLUTRR-LP.

| CoT Prompting | LMLP |
|---|---|
| Question: Wilhelmina took her uncle Hugh to the grocery store. Francisco and his brother Wesley were wrestling. Wilhelmina, Francisco's daughter, was cheering on the competition. What is the relation between Hugh and Wesley? Answer: Wesley's brother is Francisco, Francisco's daughter is Wilhelmina, Wilhelmina's uncle is Hugh. The relation of Hugh between Wesley is brother. | Task: A's aunt is B Step 1: A's sister is C Step 2: C's mother is D Step 3: D's sister is B Task: Carrie's aunt is Lynn Step 1: Carrie's sister is Natasha Step 2: Natasha's mother is Cindy Step 3: Cindy's sister is Lynn |
| Question: Constance went shoe shopping with her sister Ellen. Elsie had a daughter named Constance. Elsie had picked her daughter Margaret out the cutest new dress to wear on her birthday. Charles and his sister Kathleen have been best friends ever since childhood. Nadia and her father, James, went to the marina. James's daughter, Mabel, had purchased a boat, and they were eager to see it. Mabel bought her mother, Ellen, a puppy for her birthday. James hung his son Charles's finger paintings on the refrigerator. The paintings were right next to the paintings of Nadia, Charles's sister. Kathleen was n't old enough to make any paintings for her father, James. What is the relation between Margaret and Charles? Answer: charles' mother is nancy, nancy's daughter is elizabeth, elizabeth's husband is john, john's wife is mary, mary's brother is george, david's nephews are william, william's nephews are robert.✗ | Task: Charles's aunt is Margaret Step 1: Charles's sister is Nadia Step 2: Nadia's aunt is Margaret ✓ |

Prompts Ensembling. Table 8 shows the results of different prompts for the same task. We can see the influence of prompts on the generated proof path. The first few proof steps are largely similar to the provided example. If the provided example supplies a wrong direction, the proof is likely to be wrong. To study and exploit the benefit brought by different prompts, during experiments, we propose to use $K$ prompts alternatively for one task, where one task is marked to be successfully proved if any of these $K$ prompts gets the right result. Namely, a larger $K$ means that we have a higher probability of picking a good prompt. The default hyper-parameters $N, K$ are set to one.

**Evaluation metrics** used in this work is the **proven accuracy or success rate**. For example, for the target "Task: palau locatedIn oceania", we begin with entity "palau" and select facts from the $\mathcal{F}$. If the chosen triplet ends with entity "oceania", the proven path is correct, e.g., "micronesia locatedIn oceania" in Table 1. For LMLP, if there is no chosen triplet ends with entity "oceania", the prediction is incorrect. For CoT, we can easily compare the final answer with the ground truth.

For every query set, we calculate the average proven success rate (Number of correct proven paths/number of test samples). For the CLUTRR dataset, there are a large number of noisy triplets. For example, the target in Table 1 is "Ashley's son is Nicholas" while "Ashley's husband is Nicholas" is in $\mathcal{F}$. Once the latter rule is chosen as one proof step, the proving process will terminate because "Ashley's husband is Nicholas" ends with the entity "Nicholas", however, the proof path is wrong and

Table 4: Ablation of LMLP on CLUTRR-LP.

| Test Reasoning Length | K=1 | K=3 | K=5 | K=10 | Avg |
|---|---|---|---|---|---|
| 5 Hops | 0.3946 | 0.6865 | 0.7838 | 1.0000 | 0.7162 |
| 6 Hops | 0.5048 | 0.7143 | 0.7619 | 1.0000 | 0.7452 |
| 7 Hops | 0.4323 | 0.8065 | 0.8774 | 1.0000 | 0.7790 |
| 8 Hops | 0.5037 | 0.8000 | 0.8593 | 1.0000 | 0.7907 |
| 9 Hops | 0.3710 | 0.6452 | 0.7500 | 1.0000 | 0.6915 |
| 10 Hops | 0.5328 | 0.8279 | 0.8525 | 0.9180 | 0.7828 |

cannot induce the relation "son" between "Ashley" and "Nicholas". To evaluate the correctness of generated proof path, **human evaluations** are conducted. For each proof path, we ask annotators to answer "Yes" or "No" to whether the target relation can be induced from the proof path.

### C.1 Comparisons of LMLP and CoT

The goal of this part is to systematically compare LMLP with CoT both quantitatively and qualitatively on SRL tasks to better understand the reasoning of LMs using in-context learning.

In Figure 1, we compare LMLP to CoT and the reported performances are all human evaluation results. Qualitatively, CoT can get positive results on some query examples, for example, in Table 11, we showcase two examples where CoT can generate a correct proof path and predict the target relation at the same time. However, compared to LMLP, CoT attains inferior results on all query sets with test reasoning length $5, 6, 7, 8, 9, 10$ even using GPT-3 for text generation. Besides, as the reasoning length increases, the performance of CoT shows a clear downward trend. Table 11 shows two negative examples, where the story contains sophisticated relations and the model cannot get the right reasoning path or just generate a wrong relation. In contrast, LMLP can consistently achieves a high proven success rate (Table 2) and human evaluation score (Table 6 in Appendix), which again verifies the systematic generalization capability of LMLP. Table 3 shows examples with the same task but processed by the two methods respectively, where CoT cannot get deduce a right relation path from *Margaret* to *Charles* but LMLP can extract a simple yet right relation path. The reason why LMLP is better than CoT can be that, although CoT decomposes complex multi-hop relation reasoning tasks into a multi-step reasoning process and then predict the final results, the proof path is all generated by LMs at once. The decomposition of LMLP to multi-hop reasoning tasks is more thorough, where the generation of a proof path is divided into multi-steps and each step will be projected into the KB, which is a much stronger inductive bias. Therefore, the decomposed tasks in each step are easier to solve and the knowledge in the KB can be well exploited.

### C.2 Data Generation.

**CLUTRR-LP.** CLUTRR has 9 subsets with difference story length, named $l_2, l_3, \ldots, l_{10}$. Following [31], we convert $l_2, l_3, l_4$ to the $\mathcal{R}$ and use $l_5, \ldots, l_{10}$ to the **query sets**. As illustrated in Table. 1, data samples in CLUTRR consist of a story and a target, where the target contains two entities and the relation that is needed to be inferred, the story contains available triplets. Each sample in the $l_2, l_3, l_4$ will be converted to the format "Task: ..., Step i: ..." and added to the $\mathcal{R}$. Note that all examples in the $\mathcal{R}$ have a story length of less than five, which enables us to test the systematic generalization ability of LMLP. For CLUTRR, the story triplets in the $\mathcal{R}$ are not useful for test target proving, because they are all from different relation graphs. For example, story triplets in the $l_2, l_3, l_4$ contain "(William's brother is Steve)" while one test story on $l_5$ contains "(William's uncle is Steve)". During the evaluation, if the model chooses "(William's brother is Steve)", the proof path will be wrong. However, the similarity of these two triplets is high, the model is then easy to make errors and these noisy facts increase proof difficulties. We hence evaluate our methods in two settings considering the number of noisy facts. The simplest setting (**Test Facts Setting**) is that, when queries are from $l_i, i \in [5, \ldots, 10]$, the $\mathcal{F}$ only contains facts in $l_i$. In this case, the $\mathcal{F}_{5\sim10}$ have 251,222,275,279,285,304 facts respectively. The most difficult setting is termed **All Facts Setting**. We first extract facts in the $\mathcal{F}$ with length $l_2, l_3, l_4$ and get totally $5,210$ facts. When queries are from $l_i, i \in [5, \ldots, 10]$, the $\mathcal{F}$ contains triplets in $l_i, l_2, l_3, l_4$, where the additional $5,210$ facts are not useful for the proof path and are noisy facts. The All Facts Setting is set as our default setting and experimental results of the Test Facts Setting are mainly in the Appendix. For CoT, the $\mathcal{F}$ is needless

and the construction of prompt examples is slightly different from the procedure above. Specifically, as shown in Figure 4(b), for each target in the training samples, we need to preserve the story and extract a proof path for the target.

**Countries-LP.** Training samples in Countries are triplets that describe the *neighbor of* relation or *located in* relation of two regions/subregions/countries and can thus be directly used as $\mathcal{F}$. Because the three tasks (*S1, S2, S3*) [31] have different training sets and thus have different $\mathcal{F}$. Test samples in Countries are also triplets with specific entities and relations, hence the **query set** is just the test set of the original Countries dataset. One main difficulty in applying the proposed method to Countries is the lack of off-the-shelf proof paths ($\mathcal{R}$). The CTP [31] model is trained and used for proving each triplet in the training set. CTP returns the scores of the possible proof path and the proof with the maximum score is iteratively searched as added into the $\mathcal{R}$. After that, $924, 906, 705$ available examples are found for *S1, S2, S3* tasks respectively.

### C.3 Analysis of LMLP

Given the above observations that LMLP outperforms CoT by a large margin, we systematically analyze LMLP with extensive experiments below.

**Ablation Studies on prompting strategies.** As illustrated in Table 2, **No Prompt** means that we only feed the target directly and generate each step, prompts in the **Only Rule** baseline is one proof example with entities replaced by some symbols. We also compare LMLP to Language Planner [15], which first finds the most similar target in the $\mathcal{R}$ and uses such an example as the prompt. **LMLP-reverse** swaps the position of the abstract logic rule and its grounded example in the prompt of LMLP. For example, in Figure 3, the in-context prompt of LMLP-reverse will place `Sister(George, Nancy) ← Brother(George, Dale) ∧ Sister(Dale, Nancy)` before its abstract logic rule `Sister(A,C) ← Brother(A,B) ∧ Sister(B,C)`. Examples for all baselines are shown in Appendix Table 9.

Table 2 shows that directly applying Language Planner for relational reasoning does not work and using only facts or no prompt attain inferior performance. The possible reason for the inferior performance of Planner can be that it finds the example from $\mathcal{R}$ with the most similar task as the prompt, which usually retrieves rules with the same entities of the goal task. However, for reasoning tasks over KBs, relation contains much more information of the task than the entity. As shown in Table 9, for the task "Patricia's uncle is Donald", Planner finds the example with task "David's nephew is Don", whose following proofs do not make sense for the relation "uncle". LMLP in contrast finds an example whose task has the same relation as the goal predicate, which is more informative.

Table 5: Results of LMLP on Countries-LP. *S1, S2, S3* [31] are three different tasks with different $\mathcal{F}$ (See experimental setting for the details).

| | K=1 | K=3 | K=5 | K=10 | A Long Example |
|---|---|---|---|---|---|
| S1 | 0.7083 | 0.9583 | 1.0000 | 1.0000 | Task: A locatedIn C
Step 1: A neighborOf B
Step 2: B locatedIn C |
| S2 | 0.5000 | 0.8750 | 0.9583 | 1.0000 | Task: uruguay locatedIn south_america
Step 1: uruguay neighborOf argentina
Step 2: argentina locatedIn south_america |
| S3 | 0.7500 | 0.9167 | 0.9167 | 1.0000 | Task: sudan locatedIn africa
Step 1: sudan neighborOf central african republic
Step 2: central african republic neighborOf chad
Step 3: chad neighborOf south sudan
Step 4: south sudan neighborOf dr congo
Step 5: dr congo neighborOf republic of the congo
Step 6: republic of the congo locatedIn middle africa
Step 7: middle africa locatedIn africa |

**Effects of** $K$**.** We show evaluation results on CLUTRR-LP in Table 4 and the proposed method can generate realistic and correct proof paths. A large $K$ can further boost the performance, which also verifies the importance of prompts ensembling: Table 5 in the Appendix shows the performance on Countries-LP where almost all the query samples can be proved correctly with a large $K$. One interesting phenomenon is that LMLP can generate a much longer proof path even though the proof

path length in the rule set $\mathcal{R}$ is less than 3. This manifests a potential improvement with respect to the significant weakness in systematic generalization of fine-tuning or re-training of LMs [46]. The $\mathcal{R}$ of CLUTRR-LP contains only examples whose proof paths are less than five. However, during testing, our model can produce proof paths much longer than five steps and perform well on all query sets. **Effects of $N$.** Recall that $N$ denotes the amount of in-context examples used in one proof task. Table 7 shows an example when $N = 2$. Figure 5(a) gives ablation results, where a larger $N$ can bring consistent performance gains. However, longer prompts require larger GPU memory and there is a trade-off between memory and performance.

**Robustness to noisy facts.** As elaborated in Appendix C.2, we have 5210 noisy facts in total. Figure 5(b) shows the results when we vary the number of noisy facts, where the noisy rate is 0.5 means that we add $5210 * 0.5$ facts to the $\mathcal{F}$ during evaluation. Even though when all the 5,210 noisy facts are added to the $\mathcal{F}$ (251, 222, 275, 279, 285, 304 facts for the six sets of facts $\mathcal{F}_{5\sim10}$), namely more than $95\%$ facts are noisy, the performance is still favorable.

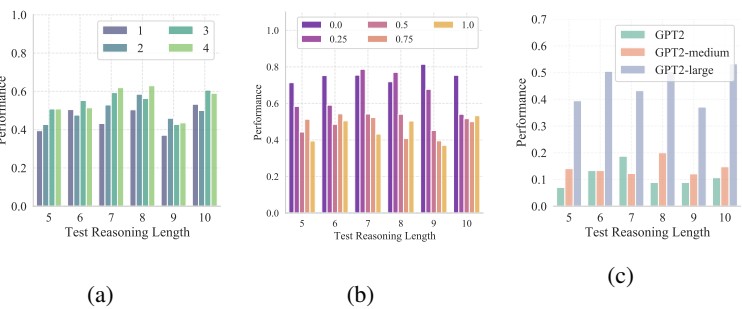

(a)                      (b)                    (c)

Figure 5: (a) Effect of the number of templates for LMLP on CLUTRR-LP. (b) The effects of noisy facts for LMLP on CLUTRR-LP. Ablation on the scaling of (c) Planning LMs.

### C.4 Analysis of Demonstrations of In-context Learning

Besides results in Table 3, we conduct additional qualitative analysis of demonstrations of in-context learning.

**Failure cases analysis of baselines.** Since the generated sentences are closely related to the prompt, Table 10 in Appendix shows that if we randomly choose prompts, the generated proof path has relations similar to the prompt but is wrong for the given task. For entity-based prompts, since the task has the same start entity as the in-context exemplar, hence the generated step 1 in this setting are very similar, leading to many wrong proof paths. Language Planner, without chain rule constraint, the generated triplets are chaos, e.g., in Example 1, the generated proof does even not contain the subject "Jon" and thus exactly wrong. Although the proposed LMLP attains a high success rate, there are also some failure cases. As shown in Appendix Table 8, an appropriate prompt needs to be chosen for the right proof paths.

Human evaluation results are shown in Appendix Table 6. When the $\mathcal{F}$ is small and without many noisy facts, almost all ablations attain high accuracy besides the language planner. Namely, the language planner either fails to prove the target task or the proof path is wrong. As we increase the number of noisy facts, the proposed LMLP shows high robustness and attains higher accuracy than other ablations.

**Takeaways.** Similar to previous works [25, 29], we find that in-context learning performance varies greatly with choices of exemplars (Table 4). One of the key findings in [29] is that even without any labeled data, LMs can achieve k-shot performance by simply prompting with demonstrations containing unlabeled inputs. Our findings are generally in-line is in line with the importance of input-label formats highlighted in the work. However, we show in Table 7 and 9 that the correct mapping of rule-example pairs are important since given only rules with symbols like $X, Y, Z$ rather than concrete entities like $China$ makes LMLP fail catastrophically. We leave explorations about zero-shot baselines of symbolic reasoning over KBs as future work.

Table 6: Human evaluation of the proposed technique. A 0.9 accuracy during human evaluation means that 90% of the model's successful proof paths in Table 2 are correct and others give incorrect reasoning paths even if they converge to the target entity. Hence the true proven success rate of LMLP in the query set with Test Reasoning Length 5 is 0.3946×0.8356.

| Test Reasoning Length | Planner | No Prompt | Only Rule | Random | Entity-based | LMLP-reverse | LMLP |
|---|---|---|---|---|---|---|---|
| 5 | 0.4615 | 0.7368 | 0.7143 | 0.7941 | 0.7872 | **0.9200** | 0.8356 |
| 6 | 0.6129 | 0.5417 | 0.5926 | 0.5366 | 0.5000 | **0.6531** | 0.4906 |
| 7 | 0.7292 | 0.6458 | 0.6471 | 0.5217 | 0.5625 | **0.8406** | 0.5970 |
| 8 | 0.4375 | 0.6977 | 0.6923 | 0.7368 | 0.7442 | **0.7619** | 0.7059 |
| 9 | 0.5417 | 0.7500 | 0.8182 | 0.7407 | 0.7419 | **0.8462** | 0.8043 |
| 10 | 0.5556 | 0.8750 | 0.7647 | 0.8421 | 0.8684 | **0.9846** | 0.8923 |

Table 7: Sensitivity analysis of prompt formats using LMLP in the Countries dataset [42].

| Method | Prompt | Output |
|---|---|---|
| Facts w/o Provenance | Task: Kazakhstan locatedIn Asia | Step 1: Kazakhstan locatedIn CentralAsia 
 Step 2: CentralAsia locatedIn Asia |
| Facts w/ Provenance | Task: Kazakhstan locatedIn CentralAsia 
 Step 1: Kazakhstan neighborOf Uzbekistan 
 Step 2: Uzbekistan locatedIn CentralAsia 
 Task: Kazakhstan locatedIn Asia | Step 1: Kazakhstan neighborOf China 
 Step 2: China neighborOf India 
 Step 3: India locatedIn SouthernAsia 
 Step 4: SouthernAsia locatedIn Asia |
| LMLP (N=1) | Task: X is located in Z 
 Step 1: X is located in Y 
 Step 2: Y is located in M 
 Step 3: M is located in Z 
 Task: Kazakhstan locatedIn Asia | Step 1: Kazakhstan neighborOf kyrgyzstan 
 Step 2: Kyrgyzstan neighborOf Uzbekistan 
 Step 3: Uzbekistan neighborOf Afghanistan 
 Step 4: Afghanistan neighborOf china 
 Step 5: China neighborOf India 
 Step 6: India neighborOf Nepal 
 Step 7: Nepal locatedIn Asia |
| | Task: X locatedIn Y 
 Step 1: X neighborOf M 
 Step 2: M neighborOf N 
 Step 3: N locatedIn Y 
 Task: Kazakhstan locatedIn Asia | Step 1: Kazakhstan locatedIn CentralAsia 
 Step 2: CentralAsia locatedIn Asia |
| | Task: X locatedIn Y 
 Step 1: X neighborOf M 
 Step 2: M locatedIn N 
 Step 3: N locatedIn Y 
 Task: Kazakhstan locatedIn Asia | Step 1: Kazakhstan locatedIn CentralAsia 
 Step 2: CentralAsia locatedIn Asia |
| | Task: X locatedIn Y 
 Step 1: X neighborOf Z 
 Step 2: Z locatedIn Y 
 Task: Kazakhstan locatedIn Asia | Step 1: Kazakhstan neighborOf kyrgyzstan 
 Step 2: Kyrgyzstan neighborOf Uzbekistan 
 Step 3: Uzbekistan neighborOf Afghanistan 
 Step 4: Afghanistan neighborOf Tajikistan 
 Step 5: Tajikistan neighborOf China 
 Step 6: China neighborOf India 
 Step 7: India neighborOf Nepal 
 Step 8: Nepal locatedIn Asia |

Table 8: Failure cases analysis - the results of different prompts for the same task.

| Method | Prompt | Output |
|---|---|---|
| No Prompt | Task: Richard's father is Donald | Step 1: Richard's grandmother is Elizabeth
Step 2: Elizabeth's son is Donald |
| Random Prompt | Task: C's father is B
Step 1: C's brother is A
Step 2: A's father is B
Task: Janet's father is George
Step 1: Janet's brother is John
Step 2: John's father is George
Task: Pat's grandfather is Jose | Step 1: Pat's uncle is Samuel
Step 2: Samuel's father is Jose |
| Rule Only | Task: B's father is C
Step 1: B's mother is D
Step 2: D's son is A
Step 3: A's father is C
Task: Richard's father is Donald | Step 1: Richard's grandmother is Elizabeth
Step 2: Elizabeth's son is Donald |
| Entity-based Prompt | Task: C's brother is A
Step 1: C's brother is B
Step 2: B's brother is A
Task: Milton's brother is Glen
Step 1: Milton's brother is Michael
Step 2: Michael's brother is Glen
Task: Milton's mother is Antonia | Step 1: Milton's daughter is Margaretta
Step 2: Margaretta's grandmother is Antonia |
| Language Planner | Task: David's nephew is Don ✗
Step 1: David's mother is Frances
Step 2: Frances's daughter is Lynn
Step 3: Lynn's son is Don
Task: Patricia's uncle is Donald | Step 1: Richard's father is Donald |
| LMLP (N=1) | Task: B's father is C
Step 1: B's mother is D
Step 2: D's daughter is A
Step 3: A's father is C
Task: Bobby's father is Hugh
Step 1: Bobby's mother is David
Step 2: David's daughter is Marie
Step 3: Marie's father is Hugh
Task: Richard's father is Donald | Step 1: Richard's mother is Maria
Step 2: Maria's husband is Donald |
| LMLP (N=2) | Task: E's father is B
Step 1: E's daughter is A
Step 2: A's sister is D
Step 3: D's brother is C
Step 4: C's grandfather is B
Task: Louis's father is Harold
Step 1: Louis's daughter is Jennifer
Step 2: Jennifer's sister is Nettie
Step 3: Nettie's brother is Samuel
Step 4: Samuel's grandfather is Harold
Task: D's father is C
Step 1: D's husband is A
Step 2: A's son is B
Step 3: B's grandfather is C
Task: Lena's father is James
Step 1: Lena's husband is Wesley
Step 2: Wesley's son is Theodore
Step 3: Theodore's grandfather is James
Task: Richard's father is Donald | Step 1: Richard's grandmother is Elizabeth
Step 2: Elizabeth's son is Donald |

Table 9: The comparison of different demonstrations on results.

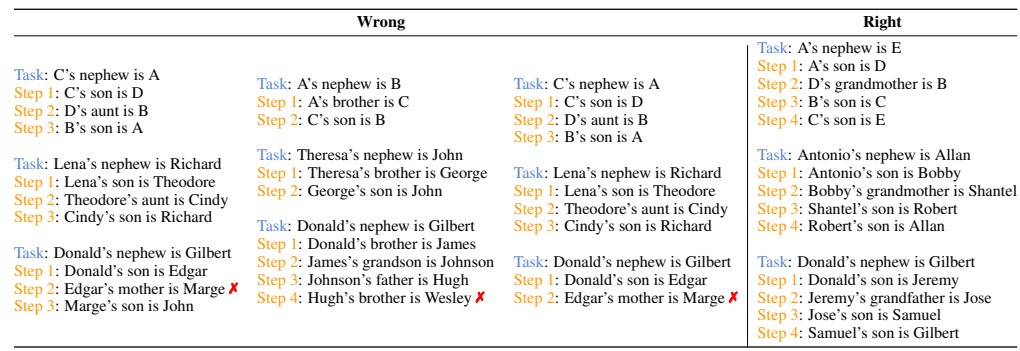

Table 10: Failure cases analysis of different baselines. Examples for each method are all from the same setting.

| Method | Example 1 | Example 2 |
|---|---|---|
| Random Prompt | Task: D's mother is B
Step 1: D's father is E
Step 2: E's daughter is C
Step 3: C's brother is A
Step 4: A's mother is B
Task: Allan's mother is Marie
Step 1: Allan's father is Robert
Step 2: Robert's daughter is Michelle
Step 3: Michelle's brother is Ronald
Step 4: Ronald's mother is Marie
Task: Jose's granddaughter is Margaretta
Step 1: Jose's father is William ✗
Step 2: William's niece is Margaretta | Task: A's grandson is B
Step 1: A's granddaughter is C
Step 2:C's brother is B
Task: Clarence's grandson is James
Step 1: Clarence's granddaughter is Charlotte
Step 2: Charlotte's brother is James
Task: Samuel's nephew is Charles
Step 1: Samuel's aunt is Marie ✗
Step 2: Marie's grandfather is Charles |
| Entity-based Prompt | Task: B's granddaughter is A
Step 1: B's daughter is D
Step 2: D's brother is C
Step 3: C's daughter is A
Task: James's granddaughter is Juanita
Step 1: James's daughter is David
Step 2: David's brother is Joshua
Step 3: Joshua's daughter is Juanita
Task: James's niece is Mary ✗
Step 1: James's daughter is Mary | Task: B's granddaughter is D
Step 1: B's grandson is C
Step 2: C's brother is A
Step 3: A's sister is D
Task: James's granddaughter is Andrea
Step 1: James's grandson is Thomas ✗
Step 2: Thomas's brother is Don
Step 3: Don's sister is Andrea
Task: James's nephew is Don
Step 1: James's grandson is Thomas
Step 2: Thomas's brother is Don |
| Language Planner | Task: James's father is Dale
Step 1: James's sister is Charlotte
Step 2:Charlotte's father is Dale
Task: Jon's grandfather is Daniel
Step 1: James's grandmother is Mary
Step 2: Dorothy's mother is Mary
Step 3: Mary's husband is Daniel ✗ | Task: Lynn's nephew is Michael
Step 1: Lynn's brother is Hugh
Step 2:Hugh's son is Johnson
Step 3:Johnson's brother is Michael
Task: Bonnie's nephew is Michael
Step 1: Ross's brother is Michael ✗ |
| LMLP (N=1) | Task: B's father is C
Step 1: B's son is D
Step 2: D's mother is A
Step 3: A's father is C
Task: Hugh's father is James
Step 1: Hugh's son is Bobby
Step 2: Bobby's mother is David
Step 3: David's father is James
Task: Irene's father is Milton
Step 1: Irene's grandfather is Jose
Step 2: Jose's mother is Mary ✗
Step 3: Mary's husband is Milton | Task: Task: D's nephew is A
Step 1: D's brother is B
Step 2: B's son is C
Step 3: C's brother is A
Task: Francisco's nephew is Clarence
Step 1: Francisco's brother is Joshua
Step 2: Joshua's son is Joseph
Step 3: Joseph's brother is Clarence
Task: Melanie's nephew is Charles
Step 1: Melanie's husband is William
Step 2: William's nephew is Kyle
Step 3: Kyle's father is Charles ✗ |

Table 11: Showcase of results and failure cases analysis of CoT.

**Right**

Question: Nettie's brother Paul took her to the fair when she was little. Paul also brought his brother Samuel. Shantel and husband Harold are trying to get custody of Shantel's granddaughter Nettie before she gets placed in foster care. What is the relation between Samuel and Harold?
Answer: Harold's wife is Shantel, Shantel's granddaughter is Nettie, Nettie's brother is Paul, Paul's brother is Samuel. The relation of Samuel between Harold is grandson.

Question: Chris wants to throw a surprise party for his brother Theodore. Chris's brother Charles helps pick the decorations. Frances combed her grandson Charles's hair. She wanted him to look good for his first day of school. What is the relation between Theodore and Frances?
Answer: Frances's grandson is Charles, Charles's brother is Chris, Chris's brother is Theodore. The relation of Theodore with Frances is grandson. ✔

Question: Kevin went to the park with his sister, Heidi, and his father, Wayne. The three of them had a wonderful time there. What is the relation between Wayne and Heidi?
Answer: Heidi's brother is Kevin, Kevin's father is Wayne. The relation of Wayne between Heidi is father.

Question: Harold was so excited to surprise his son, Louis, with the tickets to the playoffs. Shantel enjoyed a homemade dinner with her son Louis Tracy was making dinner for her son, Kevin. Shantel, Kevin's grandmother, stopped by, so Shantel made a bit more so they all had enough to eat. What is the relation between Harold and Tracy?
Answer: tracy's mother is shantel, tracy's father is harold. the relation of harold and tracy is father. ✔

**Wrong**

Question: Wilhelmina took her uncle Hugh to the grocery store. Francisco and his brother Wesley were wrestling. Wilhelmina, Francisco's daughter, was cheering on the competition. What is the relation between Hugh and Wesley?
Answer: Wesley's brother is Francisco, Francisco's daughter is Wilhelmina, Wilhelmina's uncle is Hugh. The relation of Hugh between Wesley is brother.

Question: Constance went shoe shopping with her sister Ellen. Elsie had a daughter named Constance. Elsie had picked her daughter Margaret out the cutest new dress to wear on her birthday. Charles and his sister Kathleen have been best friends ever since childhood. Nadia and her father, James, went to the marina. James's daughter, Mabel, had purchased a boat, and they were eager to see it. Mabel bought her mother, Ellen, a puppy for her birthday. James hung his son Charles's finger paintings on the refrigerator. The paintings were right next to the paintings of Nadia, Charles's sister. Kathleen was n't old enough to make any paintings for her father, James. What is the relation between Margaret and Charles?
Answer: charles' mother is nancy, nancy's daughter is elizabeth, elizabeth's husband is john, john's wife is mary, mary's brother is george, george's nephews are david, david's nephews are william, william's nephews are robert.✘

Question: Nicholas bought his brother Wayne a present. It was to congratulate him on becoming a father to his new son, Lorraine. What is the relation between Nicholas and Lorraine?
Answer: Lorraine's father is Wayne, Wayne's brother is Nicholas. The relation of Nicholas between Lorraine is uncle.

Question: William wanted to have a family cookout so he invited his brother James. James wanted to invite his other brother Cesar. Darryl's mother, Patrice, waited impatiently for him at the diner. Dan played basketball with his brother Eric. Darryl took his brother Eric to the baseball game with Nora's father Cesar. Cesar took his son Eric to go get nachos during the game. Patrice fixed her husband Cesar dinner and then they watched a movie they rented. Cesar rushed to the hospital to find out that his wife and already given birth to a boy and had named him Dan. Cesar was so excited to surprise his son, Eric, with the tickets to the playoffs. What is the relation between William and Nora?
Answer: nora has two brothers, one of them being william. william is the only one who knows about nora's pregnancy. he is also the only one who knew about nora's pregnancy.✘