# OpenReview forum: "The Impact of Symbolic Representations on In-context Learning for Few-shot Reasoning"
_NeurIPS.cc/2022/Workshop/nCSI — nCSI WS @ NeurIPS 2022 Poster_

### Official Review · Reviewer_P2GL · 2022-10-14
**Experimental Section seems to miss a discussion.**

**Rating:** 2
**Confidence:** 2

**Review:**

### Summary
The paper proposes a neuro-symbolic approach to solve reasoning tasks over KBs. The proposed LMLP approach uses logic rule templates and in-context learning of LMs for answering a relational query. To evaluate  LMLP, the authors conducted experiments on two datasets. In doing so, the paper aims to answer: what is the right representation for in-context samples? How does natural language explanation compared with symbolic provenance when acting as prompts? The conducted experiments show that eliciting LMs with logic rules and in-context learning, leveraging LM’s pre-trained knowledge, could be sufficient for solving reasoning tasks over KBs.


### Strength
The proposed method is well described and supported by illustrations and examples, which makes it easy to understand.

Well-written related work is compressed in the main text and extended in the Appendix.

Experiments seem to be well conducted.

### Weaknesses
However, while the experimental results are displayed in Tables 2 and 3, and further details are discussed in the Appendix, a discussion in the main text is missing.

Since I do not see other major weaknesses and the remaining parts are well written, and the experiments seem to be well conducted, I would still vote for acceptance if the authors add a discussion of the experiments similar to the ones in the appendix. I suggest moving Tables 3 and 4 to the Appendix instead.

Minor:
Line 171 Typo: We probes
The first bullet in the checklist: Section ??
what is the right representations for -> "representation" or "are"

---

### Official Review · Reviewer_BGaz · 2022-10-14
**Insightfull emperical evaluation of the novel method for investigating in-context learning setting**

**Rating:** 3
**Confidence:** 1

**Review:**

Towards progressing language models (LMs) capacity in in-context learning for reasoning, the authors propose the novel algorithm of LMLP. It uses logic rule templates and examples combined with pre-trained knowledge to do in-context learning iteratively to answer a relational query. This effectively yields a question-answering perspective to the task. Overall, LMLP consists of two LMs: Planning-LM and Translation-LM. The model is being compared with different versions of Chain-of-Thought models in experimental settings of CLUTRR-LP and Countries-LP datasets. Interestingly enough, few-shot in-context learning allows for LMs to incorporate background-knowledge without retraining. Overall, the proposed method offers an interesting perspective for further explorations to be carried out. Except for a few typos, there are no remarks to be made.

---

### Meta-Review · Area_Chair_ft8x · 2022-10-19

**Recommendation:** 2
**Confidence:** 3

**Metareview:**

This paper proposed to design a model LMLP that learns from demonstrations containing
 logic rules and  examples to iteratively reason over knowledge bases (KB).
The authors propose a LMLP approach uses logic rule templates and
in-context learning of LMs for answering a relational query. They show that LMs
can be prompted with demonstrations to generate plausible provenance for reasoning tasks over
KBs.  LMLP provides some insights towards understanding in context
learning. It also suggests a way to ground GPT that types language to non-linguistic symbols in the
KB. Overall, the paper provides useful insights and the experiments suggest that
the approach may be promising. It as a good workshop paper, with potential for
further developments.

---

### Decision · Program_Chairs · 2022-10-20

Accept (Poster)